



Manuscript prepared for Atmos. Meas. Tech.
with version 2015/11/06 7.99 Copernicus papers of the LaTeX class copernicus.cls.
Date: 9 March 2016

# Comparison of aerosol properties retrieved using GARRLiC, LIRIC, and Raman algorithms applied to multi-wavelength LIDAR and sun/sky-photometer data

V. Bovchaliuk[1], P. Goloub[1], T. Podvin[1], I. Veselovskii[2], D. Tanre[1], A. Chaikovsky[3], O. Dubovik[1], A. Mortier[4], A. Lopatin[1], M. Korenskiy[2], and S. Victori[5]

[1]Laboratoire d'Optique Atmospherique, Lille1 University, Villeneuve d'Ascq, France
[2]Physics Instrumentation Center of the General Physics Institute, Troitsk, Moscow Region, Russia
[3]Institute of Physics, NAS of Belarus, Minsk, Belarus
[4]Norwegian Meteorological Institute, Oslo, Norway
[5]Cimel advanced monitoring, Paris, France

*Correspondence to:* Valentyn Bovchaliuk (bovchaliukv@gmail.com)

**Abstract.** Aerosol particles are important and highly variable components of the terrestrial atmosphere, and they affect both air quality and climate. In order to evaluate their multiple impacts, the most important requirement is to precisely measure their characteristics. Remote sensing technologies such as LIDAR (LIght Detection And Ranging) and sun/sky-photometers are powerful tools for
determining aerosol optical and microphysical properties. In our work, we applied several methods to joint or separate LIDAR and sun/sky-photometer data to retrieve aerosol properties. The Raman technique and inversion with regularization use only LIDAR data. The LIRIC (LIdar-Radiometer Inversion Code) and recently developed GARRLiC (Generalized Aerosol Retrieval from Radiometer and LIDAR Combined data) inversion methods use joint LIDAR and sun/sky-photometer data.
This paper presents a comparison and discussion of aerosol optical properties (extinction coefficient profiles and LIDAR ratios) and microphysical properties (volume concentrations, complex refractive index values, and effective radius values) retrieved using the above-mentioned methods. The comparison showed inconsistencies in the retrieved LIDAR ratios. However, other aerosol properties were found to be generally in close agreement with the AERONET (AErosol RObotic NETwork)
products. It future studies, more cases should be analysed in order to clearly define the peculiarities in our results.

## 1   Introduction

In situ and remote sensing measurements are the two main approaches used for aerosol observations. The former involves direct measurements of particles using instruments at the survey points.
The latter involves measuring aerosol properties from a distance without direct interaction with par-





ticles. Remote sensing methods can be categorized into active and passive depending on the kind of instrument used. Instruments belonging to the passive category measure the modified solar radiation after interactions with particles. One of the most common instruments in this category, a sun/sky-photometer, measures both direct and diffuse solar radiation. These data can be used in inversion

algorithms (Dubovik and King, 2000; Dubovik et al., 2011) to retrieve several column-integrated aerosol properties such as the aerosol optical depth (AOD), single scattering albedo (SSA), particle size distribution (SD), effective radius ($r_{eff}$), and complex refractive index (CRI, including real (RRI) and imaginary (IRI) parts of refractive index). Instruments belonging to the active category of remote sensing measure scattered radiation emitted by themselves; one of the most well

regarded and widely used instrument in this category is a LIDAR (light detection and ranging). LIDARs are used for profiling atmospheric variables such as the temperature, pressure, humidity, wind speed and its direction, and the amount of trace gases and aerosols. The main advantages of LIDAR measurements include high spatial vertical resolution and applicability during night-time and in cloudy environments. Current multi-wavelength LIDAR observations can provide comprehensive

and quantitative information regarding aerosol properties (Böckmann et al., 2005; Veselovskii et al., 2015, 2016; Nicolae et al., 2013; Granados-Muñoz et al., 2014).

Several methods, techniques, and algorithms can be used to obtain the optical and microphysical characteristics of aerosols. These methods generally use different sets of data. For instance, AERONET (AErosol RObotic NETwork) inversion code uses only sun/sky-photometer data (Dubovik

and King, 2000). Similarly, the Raman technique and regularization algorithm use only LIDAR data (Ansmann et al., 1990; Weitkamp, 2005; Veselovskii et al., 2002). The LIRIC (LIdar-Radiometer Inversion Code) and GARRLiC (Generalized Aerosol Retrieval from Radiometer and LIDAR Combined data) algorithms, on the other hand, use both the sun/sky-photometer and LIDAR data (Lopatin et al., 2013; Chaikovsky et al., 2015). Because these methods use different datasets, they are applica-

ble during different observational times. For instance, while the Raman technique is most suitable for night-time observations, sun/sky-photometers do not make measurements at that time. Further, the GARRLiC algorithm, which is included in the GRASP (Generalized Retrieval of Atmosphere and Surface Properties) inversion code (Dubovik et al., 2011), can separate the fine and coarse modes of aerosols, thus, resulting in the retrieval of particle characteristics separately for both modes. While

different methods retrieve different sets of aerosol characteristics, all of them are aimed at obtaining detailed results. The objective of our study is to discriminate and compare the common aerosol characteristics obtained through different methods.

Section 2 describes the observation sites where the measurements were carried out. This section also describes a new LIDAR system, called LILAS (LIlle LIDAR AtmosphereS), which was used

at the observation sites. Section 3 presents the methods considered in our study and discusses their potential, applicability, and the common aerosol properties that were considered for comparison.





Section 4 presents three dust cases that were selected and analysed by using the algorithms described in section 3. The main conclusions and perspectives are given in the last section.

## 2 Observational sites and the LIDAR system

The LIDAR system used in this work belongs to Laboratoire d'Optique Atmospherique (LOA). This system is operated at the campus of Lille University, France. The campus area is influenced mainly by urban and industrial pollutant emissions, marine aerosols, and by mineral dust and aerosols from volcanic eruptions several times every year (Mortier et al., 2013). Other remote sensing and in situ instruments are also operational at this site. Among them is a lunar-photometer for observing AOD and

Angström exponent ($\alpha$) values on clear nights within the half moon to full moon lunar phases. LOA is a permanent LIDAR site. However, for the study of Saharan dust over West Africa (SHADOW2 campaign), a multi-wavelength LIDAR was moved to M'Bour city (Dakar site) in Senegal at the beginning of January 2015. The Dakar site is influenced by mineral dust during March–April and biomass burning during December–January. The two main objectives of the campaign were: (*i*) to

record the physical and chemical properties of aerosols over the regions impacted by considerable amounts of dust particles; (*ii*) to study the aerosol dynamics. Seven laboratories with 18 instruments took part in the campaign.

The LILAS system was assembled and setup in December 2013, and observations started in January 2014. The system is composed of a laser (Spectra-physics, INDI-40) emitting at wavelengths of

1064 nm, 532 nm, and 355 nm (100mJ/20Hz), a Newton telescope, a beam rotator, and a receiving module. The beam rotator can be used for near or far range observations by changing the overlap function. Several receiving modules were added in April 2014, and the system now consists of five elastic channels (355 nm and 532 nm both parallel and perpendicular for analog and photo-counting, and 1064 nm for total analog) and three Raman channels (387 nm for analog and photo-counting, and

408 nm and 608 nm for photo-counting). During the SHADOW2 campaign, the vibrational Raman channel at 608 nm was changed to a rotational channel at 530 nm. This rotational Raman channel showed a good and stable performance (Veselovskii et al., 2015). The system can be remotely operated and is coupled with a RADAR (RAdio Detection And Ranging) for reasons such as automatic discontinuation control and airplane safety.

The Lille site became an observation station of the European Aerosol Research LIDAR Network (EARLINET) in the summer of 2014. The main goal of the network is to provide a comprehensive, quantitative, and statistically significant database on aerosol distributions. The network has some special criteria for data quality assurance, such as a telecover test, a trigger delay, dark measurements, depolarization calibration, and regular check-ups of the Rayleigh fits (Freudenthaler, 2007, 2008,

2010; Freudenthaler et al.). LILAS has passed all the EARLINET tests and check-ups except for depolarization calibration, which is currently in progress.



## 3 Retrieval algorithms

Depending on the LIDAR characteristics, different techniques can be used for obtaining optical and microphysical properties of aerosols. All the methods and algorithms that were used for data processing are introduced in this section.

Elastic-backscatter LIDAR is considered to be a classic form of LIDAR technology (Weitkamp, 2005). This technology is based on the measurement of elastically scattered light in the backward direction. The common method that derives aerosol optical characteristics is the Klett method (Klett, 1981, 1985). This method is based on the relationship between the extinction and backscatter coefficients. The BASIC algorithm (Mortier et al., 2013) based on the Klett method has been developed at LOA and and is successfully implemented into routine for mono-wavelength LIDAR data. This algorithm retrieves an extinction coefficient profile ($\sigma_{aer}(z)$) following an iterative procedure based on a dichotomy where the LIDAR Ratio (LR) can vary in the range from 10 to 140 sr. The procedure ends when the integral of the extinction profile is close to the AOD measured by a sun/sky-photometer within $\Delta\text{AOD} = 0.01$ accuracy.

The Raman LIDAR technique is a widely known technique in the LIDAR community for obtaining aerosol optical properties ($\sigma, \beta, LR$) (Ansmann et al., 1990). This technique is based on the scattering of incident LIDAR light with photon energy shifts due to vibrational or rotational modes of the molecules. It is mostly used at night-time when the signal to noise ratio is the highest, owing to the absence of scattered sunlight by atmospheric molecules. Assuming that the aerosol extinction coefficient depends on the wavelength through $\alpha$, the former can be found calculated as (Weitkamp, 2005):

$$\sigma_{aer}(\lambda_L, z) = \frac{\dfrac{d}{dz}\left[ ln\dfrac{N(z)}{z^2 P(z)} \right] - \sigma_{mol}(\lambda_L, z) - \sigma_{mol}(\lambda_R, z)}{1 + \left( \dfrac{\lambda_L}{\lambda_R} \right)^{\alpha}} \tag{1}$$

where $P(z)$ is the power received at the Raman wavelength $\lambda_R$ from distance $z$; $N(z)$ is the molecule number density; $\sigma_{mol}(\lambda_L, z)$ and $\sigma_{mol}(\lambda_R, z)$ are the extinction coefficients due to absorption and Rayleigh scattering by atmospheric molecules for emitting LIDAR and Raman wavelengths, respectively; and $\alpha$ is the Angström exponent. The aerosol backscatter coefficient can be calculated from the ratio of the elastic signal to Raman signal by using a coefficient determined at a reference point where no aerosol is expected.

A variety of methods can be used to retrieve aerosol microphysical properties using LIDAR data. They can be divided into three main groups (Weitkamp, 2005). The methods belonging to the first group combine measurements from several instruments that provide enough information to retrieve aerosol microphysical properties. For such methods, the collocation of measurements by different instruments in space and time is necessary. The LIRIC algorithm belongs to this group; it success-



fully retrieves height-resolved aerosol optical and microphysical properties separately for fine and coarse modes (Chaikovsky et al., 2012, 2015; Wagner et al., 2013; Granados-Muñoz et al., 2014). The algorithm uses AERONET inversion products such as column volume concentration, volume-specific backscatter, and extinction coefficients as a priori information (Chaikovsky et al., 2015). The specific products include backscatter ($\beta$), extinction ($\sigma$) and volume concentration (V) profiles,

Angström exponent ($\alpha$) values, and LIDAR (LR) and depolarization ($\delta$) ratios. A deeper synergy between the LIDAR and sun/sky-photometer data is achieved in the GARRLiC algorithm developed at LOA (Lopatin et al., 2013). GARRLiC inverts the coincident LIDAR and sun/sky-photometer radiometric data simultaneously. The other marked distinction between GARRLiC and LIRIC is the inversion of two distinct aerosol modes, which makes it possible to retrieve aerosol optical and mi-

crophysical properties independently for both the fine and coarse modes. Such differences in the algorithms can influence the results obtained by the two systems significantly. The GARRLiC method is based on the Dubovik inversion code (Dubovik and King, 2000; Dubovik et al., 2011), which has been previously used for processing AERONET data. The synergistic retrieval improves aerosol retrieval properties; the LIDAR observations are expected to improve the observations of the columnar

properties of aerosols in the backscattering direction, and sun/sky-photometers provide information on aerosol properties, such as their amount or type, required for LIDAR retrievals without making assumptions based on climatological data.

GARRLIC has been designed to provide two independent vertical concentration profiles for the fine and coarse modes of aerosols, since in most cases, aerosols are believed to consist of two modes.

However, it works for single-mode inversions as well. In such cases, a single value for the total amount of particles is retrieved. The algorithm is quite flexible in this regard, single- or double-mode inversion can be chosen by the user. Further, single- or multi-wavelength LIDAR data can be used. In case of multi-wavelength LIDAR data, aerosol properties can be retrieved for fine and coarse modes separately or together for the total amount of particles. In case of single-wavelength

LIDAR data, the aerosol properties can be retrieved only for the total amount of aerosols. Depending on the different configurations of single- or double-mode inversion employed and the use of single- or multi-wavelength LIDAR data, different sets of aerosol parameters can be retrieved (see Fig. 1). Spectral information from multiple wavelengths is used to distinguish the contribution of fine and coarse aerosol modes. It should be noted that aerosol events characterized mainly by one type of

aerosols or a mixture of particles similar in size (aerosol types are not distinguished inside the mode of particles) should be retrieved by using the configuration of single mode inversion.

As for the second group of methods, optical properties ($\beta$ and $\sigma$ profiles) are calculated using Mie theory and are compared with the results obtained by using the Raman technique (Wandinger et al., 1995; Barnaba and Gobbi, 2001). In these methods, aerosol microphysical properties such as SD

and CRI are assumed as a priori information. Such methods are used in case of atmospheric layers with single, well-known type of particles. For instance, such methods can characterize the particles



of polar stratospheric clouds, volcanic ejecta, and some stratospheric particles. However, owing to the presence of a variety of particles and rapid changes in the atmospheric conditions, such methods are not applicable to the troposphere.

The third group consists of mathematical approaches that use $\beta$ and $\sigma$ coefficient profiles at multiple wavelengths (only LIDAR measurements). Such methods were developed from the methods of the second group, but they require a lower number of a priori parameters (Müller et al., 1999; Veselovskii et al., 2002, 2004; Shcherbakov, 2007). The algorithm called inversion with regularization (Regularization) developed by Veselovskii et al. (2002, 2004) has also been considered in

this work. A simplified set of LIDAR data (three backscatter (355, 532, and 1064 nm) and two extinction (355 and 532 nm) coefficients - the so called $3\beta + 2\sigma$ dataset) allows the retrieval of the main aerosol microphysical properties (Veselovskii et al., 2005). Aerosol optical properties that are required for the Regularization algorithm can be derived using the Raman technique. The main aerosol microphysical products of the Regularization algorithm are the CRI, $r_{eff}$, number, surface

area, and volume concentrations (Veselovskii et al., 2002).

These groups of retrieval methods use different types of measurements, and also, different amounts of information. For instance, while Regularization uses the $3\beta + 2\sigma$ set of optical data, AERONET uses up to ∼30 measurements (direct and diffuse almucantar measurements) at each wavelength. Hence, it is important to compare the particle properties retrieved with these methods for these dif-

ferent groups. If different algorithms retrieve similar aerosol properties, it will mean that they are in agreement and can complement each other for data processing during long-term day-night observations.

Aerosol characteristics that are common to LIRIC, GARRLiC, and Regularization algorithms are $\sigma$, LR, CRI, V, and $r_{eff}$. The challenging issue here is that no perfectly coincident measurements

exist that can be used by these algorithms. The standard Raman technique preferably uses LIDAR measurements during night-time, while the sun/sky-photometers require sunlight. Consequently, for a comparison of the retrieved aerosol properties by using the GARRLiC/LIRIC and Regularization algorithms, early morning or late evening data under stable atmospheric conditions should be selected. Three events fulfilling these requirements were selected and analysed.

## 4    Applications


Several dust events were selected from the LILAS measurements over the Lille and Dakar sites. These days had moderate (AOD $\simeq 0.5$ at 440 nm) to high (AOD $\simeq 1.5$ at 440 nm) aerosol loads. Back-trajectories (Draxler and Rolph, 2015; Rolph, 2015) and the NMMB/BSC-Dust model (Non-hydrostatic Multiscale/Barcelona Supercomputing Centre Dust model, (Pérez et al., 2011; Haustein

et al., 2012) confirmed the origin of aerosols from mineral dust and showed the source locations. In

case of local dust events, the back-trajectory analysis was not used. More details and results of the comparison of each event are presented below.

The AERONET products are presented herein for comparison. As it is used as a priori information for the LIRIC algorithm, the LR retrieved by LIRIC are presented along with the AERONET

characteristics (marked by ** in Tables 2 and 3). Mass concentration profiles can be obtained simply by multiplying the volume concentration profiles, V, with the mass density of fine and coarse mode particles. The densities of the fine and coarse modes are 1.5 $g/cm^3$ and 2.6 $g/cm^3$, respectively (Binietoglou et al., 2015; Ansmann et al., 2011, 2012; Haustein et al., 2012). This density for the coarse mode is also considered in the NMMB/BSC-Dust model.

The GARRLiC and LIRIC algorithms produce uncertainties with the retrieved aerosol properties. For the GARRLiC algorithm, systematic and random errors are presented. For the LIRIC algorithm, only the dispersion of aerosol volume concentration profiles is presented. This work presents only the uncertainties regarding the directly retrieved aerosol properties. Uncertainties on the derived aerosol properties ($\sigma$, LR, SSA profiles) are not presented owing to their high values as derived

by GARRLiC (rough estimations were about 100% and more). The uncertainties in the volume concentration profiles retrieved using the Regularization algorithm are assumed to be about 20% (Veselovskii et al., 2004, 2005).

As this work mainly deals with mineral dust with inclusions of marine aerosol particles, it will be useful to consider the particle properties obtained from previous studies. According to Weitkamp

(2005); Müller et al. (2005, 2013); Pitari et al. (2015); Dubovik et al. (2002), the typical values of $r_{eff}$ for desert dust vary within the range of 1.2–2.4 $\mu m$, and $r_{eff}$ for the coarse mode of sea salt is close to 2.7 $\mu m$ (Dubovik et al., 2002). The SSA for dust particles increases from 0.80 to 0.99 in the ultraviolet–near-infrared range (Collaud Coen et al., 2004; Dubovik et al., 2002). The SSA for marine aerosols is high, at ~0.98, and the value remains stable at all wavelengths. The RRI varies

from 1.5 to 1.6 for dust particles and is close to 1.36 for marine particles. The IRI decreases from 0.02 to 0.001 in the ultraviolet–near-infrared range for dust particles, and is close to 0.001 for marine particles. For Saharan dust, the LR varies within the range of 50–80 sr at a wavelength of 532 nm, and it is significantly lower, at 20–35 sr, for marine particles (Weitkamp, 2005; Müller et al., 2007, 2010; Groß et al., 2011). The depolarization ratio is high, being close to 30–35% for dust particles,

whereas marine particles have a significantly lower $\delta$, i.e. close to 5% (Freudenthaler et al., 2009; Groß et al., 2011).

### 4.1 Analysis of a moderate dust event in Lille on 30 March 2014

The dust event detected over Lille on 30 March 2014 was characterized as moderate in terms of dust intensity, but heavy in terms of the aerosol load for Lille site (AOD 440 nm $\approx 0.52$; $\alpha \approx 0.27$ for

440/870 nm). The back-trajectory analysis showed that aerosols, which were located in the altitude range of 3 to 6 km, had their origin in the Saharan region (Fig. 2), and aerosols located up to 2



km travelled from south and south-east France. According to LIDAR measurements, very thin and homogeneous cirrus clouds with negligible effect on AOD were present at 11 km. Cross-examination was done using almucantar sky radiance measurements in order to prevent cloud contamination. The

relative deviation between the left/right sky radiance measurements in almucantar geometry was found to be less than 20%. Cirrus clouds were neither detected by us nor by AERONET criteria (Holben et al., 2006); the exact time of the sun/sky-photometer measurements was 7:42 UTC. The NMMB/BSC-Dust model (operated by the Barcelona Supercomputing Center, www.bsc.es/projects/earthscience/NMMB-BSC-DUST/) confirmed dust emissions over Algeria that travelled towards

Lille (Fig. 3).

The configuration of LILAS was changed from 3 channels (355 nm parallel and perpendicular, and 532 nm total) to 8 channels (355 and 532 nm parallel and perpendicular, and 387, 408, 608, and 1064 nm total) in April 2014. Hence, the Saharan dust event could not be analysed by the Raman and Regularization algorithms. The results of the LIRIC inversion using only the data from 355 and

532 nm wavelengths were found to be rather unsatisfactory for the coarse mode (Chaikovsky et al., 2015). Consequently, only the GARRLiC and BASIC algorithms were considered in our analysis. The LIDAR elevation angle during the measurements was $56°$.

Aerosol properties retrieved by the GARRLiC and BASIC algorithms and AERONET products are presented in Table 1 and Figs. 4 - 7. The columnar-integrated GARLLiC SSA values increase

with the wavelength, i.e. from 0.95 at 355 nm to 0.98 at 1020 nm, while the SSA for the fine mode remains stable, i.e. equal to 0.98 at all wavelengths, the SSA of the coarse mode increases from 0.91 to 0.98. The RRI is close to 1.54 for the coarse mode and 1.46 for the fine mode particles. The IRI decreases for the coarse mode of particles, and remains stable for the fine mode. The CRI values retrieved by GARRLiC are in agreement with the AERONET retrievals. The GARRLiC LR values

are lower in comparison to the ones retrieved by AERONET at wavelengths of 355, 440, and 532 nm, while they are almost equal at others. While the BASIC LR values at 532 nm are close to the values interpolated by AERONET values, the values at 355 nm are much lower than the ones retrieved by GARRLiC and AERONET. The effective radius for the coarse mode of particles is high and is close to 2.1 $\mu m$, and the $r_{eff}$ for the fine mode is close to the value of urban-industrial particles.

The size distribution (see Fig. 4) clearly shows the predominance of coarse mode particles with two maxima. The first one with lower radii likely indicates dust particles, and the second one with larger radii refers to the particles of thin cirrus clouds. The sphericity parameter retrieved by GARRLiC is in agreement with the one from AERONET, both being close to 1%. The volume concentration profile (Fig. 5) clearly shows an increase in the volume of coarse mode particles, and the

extinction Ångström exponent ($\alpha^{ext}$) profile at wavelengths of 355 and 532 nm clearly shows a dust layer ($\alpha^{ext} \approx 0.2$) in the 3–6 km altitudinal range. The extinction profiles retrieved by BASIC and GARRLiC (Fig. 6) show some differences because the BASIC algorithm derives columnar-integrated values of LR, whereas the values of LR retrieved by GARRLiC change with altitude (Fig.



**Table 1.** Aerosol properties retrieved by GARRLiC, BASIC, and AERONET. The LR values marked by **
were linearly interpolated/extrapolated to LIDAR wavelengths. The abbreviations f, c, and t corresponds to
fine, coarse, and total aerosol modes, respectively (AOD 440 nm $\approx 0.52$; $\alpha \approx 0.27$ for 440/870 nm).

| $\lambda$ [nm] | GARRLiC | | | | | | | BASIC | AERONET | | |
|---|---|---|---|---|---|---|---|---|---|---|---|
| | $r_{eff}$ [$\mu m$] | Sph % | RRIf | RRIc | IRIf | IRIc | LR [sr] | LR [sr] | RRI | IRI | LR [sr] |
| 355 | | | 1.47 | 1.54 | 0.002 | 0.002 | 53 | 25 | | | 63** |
| 440 | f: 0.1 | | 1.46 | 1.54 | 0.002 | 0.002 | 48 | | 1.48 | 0.002 | 57 |
| 532 | c: 2.1 | 1% | 1.46 | 1.53 | 0.002 | 0.002 | 45 | 53 | | | 52** |
| 675 | t: 1.1 | | 1.47 | 1.54 | 0.002 | 0.001 | 42 | | 1.52 | 0.001 | 43 |
| 870 | | | 1.46 | 1.54 | 0.002 | 0.001 | 41 | | 1.51 | 0.001 | 43 |
| 1020 | | | 1.47 | 1.54 | 0.002 | 0.001 | 42 | | 1.51 | 0.001 | 43 |

7). The extinction profile of the coarse mode of aerosols clearly shows a dust layer. It should be
noted at this point that the BASIC algorithm works based on mono-wavelength LIDAR signals. In
case of multi-wavelength LIDAR measurements, BASIC retrieves aerosol properties individually at
each wavelength. The LR value obtained at 355 nm by the BASIC algorithm is quite low; there-
fore, the $\sigma$ profile at this wavelength is not presented. The SSA profiles decrease in the dust layer at
both 355 nm and 532 nm; however, it is interesting to note that the SSA at 355 nm decreases more
significantly than the SSA at 532 nm (Fig. 7).

The retrieved aerosol properties indicate two layers. The lower layer has high SSA, LR, and $\alpha^{ext}$
values, which are typical for urban-industrial particles. Meanwhile, the upper layer has a predomi-
nance of aerosol particles in the coarse modes and the contribution of the fine mode remains high
(see $\sigma$ for the fine mode on the left panel in Fig. 6). This mixture of dust and some fine particles
results in lower LR and higher SSA values than for aerosols from mineral dust only (Balis et al.,
2004; Giannakaki et al., 2010; Petzold et al., 2011). Unfortunately, no mass concentration profiles
could be obtained by the NMMB/BSC-Dust model.

### 4.2 Analysis of a heavy dust event in Dakar on 29 March 2015

The second event considered in this work was also a dust event, but it occurred over the Dakar
site during the SHADOW2 campaign. Three time ranges were selected for the analysis. Day-time
data from 15:50 to 19:00 were selected for the Raman technique. For the GARRLiC and LIRIC
algorithms, LIDAR signals were averaged for 20 min at the time of measurement by the sun/sky-
photometer (16:49 UTC). A third data range was selected for the Regularization and Raman methods
from 23:30 to 01:10 during night-time measurements. All aerosols were found in the boundary layer



290   for all time ranges. During the day-time measurements, the altitude of the boundary layer was 2.5
km, and it came down to 2 km at night. The day-time event was characterized by a high aerosol
load (AOD 440 nm $\approx 1.35 \pm 0.20$; $\alpha \approx -0.04 \pm 0.01$ for 440/870 nm), and the night-time event was
characterized by a lower aerosol load (AOD 440 nm $\approx 0.83 \pm 0.03$; $\alpha \approx 0.08 \pm 0.02$ for 440/870
nm). The NMMB/BSC-Dust model showed a local dust event over the Dakar site with an AOD

295   range of 0.8–1.6 at the 550 nm wavelength (Fig. 8) for both day- and night-time measurements. A
wind LIDAR instrument was installed on the site during the SHADOW2 campaign (Wang et al.,
2014), and it captured vertically resolved wind speeds and the direction of wind at the site for up to
2 km. The LIDAR data showed the wind direction to be north–north-east with a speed of 5 to 10 m/s
in the full altitudinal range for the day-time measurements; the wind direction was north-east with

300   a speed of 10 to 15 m/s in the altitudinal range of up to 1.5 km, and lower speeds of 5 to 10 m/s
were present in upper altitudes for the night-time measurements. Therefore, while presumably the
atmospheric conditions in terms of aerosol types should have remained the same during the event,
the aerosol load decreased over the day- to night-time measurement timeframe. Also, the presence of
marine particles were not expected because of the lower wind speeds and their northwardly direction

305   during the day.

The aerosol properties retrieved by the GARRLiC and Regularization algorithms for the day- and
night-time measurements, respectively, are presented in Table 2.

**Table 2.** Aerosol properties during the dust event over the Dakar site on 29 March 2015. Here and further,
the LR values marked by ** were retrieved by using the LIRIC algorithm. Only the values given for all the
wavelengths refer to the column-integrated property. Day: AOD 440 nm $\approx 1.35 \pm 0.20$; $\alpha \approx -0.04 \pm 0.01$.
Night: AOD 440 nm $\approx 0.83 \pm 0.03$; $\alpha \approx 0.08 \pm 0.02$.

| $\lambda\,[nm]$ | GARRLiC | | | | | AERO-NET | Raman (Day) | Raman + Regularization (Night) | | | |
|---|---|---|---|---|---|---|---|---|---|---|---|
| | $r_{eff}$ [$\mu m$] | Sph % | RRI | IRI | LR [sr] | LR [sr] | LR [sr] | $r_{eff}$ [$\mu m$] | RRI | IRI | LR [sr] |
| 355 | | | 1.59 | 0.003 | 37 | 82** | ~57 | | | | ~70 |
| 440 | | | 1.59 | 0.003 | 33 | 74 | | | | | |
| 532 | | | 1.59 | 0.002 | 28 | 58** | ~53 | | | | ~58 |
| 675 | 1.9 | 20% | 1.58 | 0.002 | 25 | 43 | | 1.1 | 1.53 | 0.010 | |
| 870 | | | 1.57 | 0.002 | 24 | 37 | | | | | |
| 1020 | | | 1.56 | 0.002 | 22 | 35 | | | | | |
| 1064 | | | 1.56 | 0.002 | 22 | 34** | | | | | |

Single-mode GARRLiC inversions were considered and performed in this work because of the
huge predominance of coarse mode particles. The effective radius value is high and close to 1.9





during the day-time, and decreases to 1.1 $\mu m$ at night. The RRI values are high, being close to 1.58 ± 0.02 during the day-time measurements; then, values become lower and close to 1.53 ± 0.05 at night. The IRI values decrease from 0.003 to 0.002 in the UV–near-infrared range during the day-time, and are higher at all wavelengths and close to 0.010 ± 0.005 at night. The higher effective radius and lower values of IRI during the day-time measurements can be explained by the contribution of non-absorbing marine particles. For both the day and night cases, the Angström exponent is close to 0. Regarding absorption, the SSA values obtained by GARRLiC increase from 0.87 to 0.97 in the UV–near-infrared range. The day-time LR values are similar at 532 nm, whereas the ones retrieved by GARRLiC are much lower. The LR values at 355 nm during the day-time measurements differ for all the algorithms, being close to 57, 82, and 37 sr for the Raman, LIRIC, and GARRLiC algorithms, respectively. The Raman LR values slightly increase from ∼53 to ∼58 sr at 532 nm, and it significantly increases form ∼57 to ∼70 sr at 355 nm over the day- to night-time measurement timeframe. Such a behaviour could be explained by the influence of marine aerosols during day-time.

It was observed that the IRI, SSA, SD, and $r_{eff}$ retrieved by GARRLiC were in good agreement with AERONET products. However, RRI values and parameter of particle sphericity differed. While the AERONET RRI is equal to 1.53 and the sphericity is equal to 0%, the RRI retrieved by GAR-RLiC is close to 1.58 and the sphericity is ∼20%. The differences in the LR values are presented in Table 2 and are discussed above.

Figure 9 shows that the SD values obtained from GARRLiC and AERONET are in good agreement. Figure 10 presents the aerosol volume concentrations, V, retrieved with the GARRLiC, LIRIC, and Regularization Algorithms. Because of the use of single-mode inversion by GARRLiC, only the overall V profile was obtained; however, the LIRIC algorithm provided both fine and coarse mode volume concentrations. Because of a high background noise, the Regularization algorithm was not applied to day-time measurements; only night-time V is presented with this algorithm. The GAR-RLiC and LIRIC volume concentrations are in good agreement. Unfortunately, a comparison between the day- and night-time V values was not possible because of a significant decrease in the AOD values. The relative uncertainty in V obtained from the Regularization method was expected to be about 20%; the GARRLiC and LIRIC uncertainties are plotted in Fig. 10.

The extinction profiles (Fig. 11) at all the wavelengths were found to be in reasonable agreement. The night-time values of $\sigma$ are lower in accordance with the lower AOD values. The top boundary of the dust layer decreases from 2.5 km during the day to 2 km at night. The GARRLiC extinction pro-files are much smoother because LIDAR signals were averaged into 60 points during the data prepa-ration phase. The day-time Raman LR values (Fig. 12) decrease with altitude and, therefore, result is slightly correct $\sigma$ profiles; however, GARRLiC and LIRIC retrieved only the column-integrated LR in this case (GARRLiC retrieved vertically resolved LR in case of fine and coarse modes in-version). As mentioned above, lower day-time LR at 355 nm can be explained by the presence of





marine aerosols. The particle depolarization, presented in Fig. 12, is lower during day-time (close to 29%) and higher at night-time when no marine particles are expected (close to 34%). These LR and particle depolarization values are common for mineral dust, especially at night-time. The $r_{eff}$

profile retrieved by Regularization is close to 1.1 $\mu m$ at the 0.9–1.6 km altitudinal range, and higher values up to 1.4 $\mu m$ were observed below 0.9 km, whereas lower values close to 0.8 were observed above 1.6 km. The Regularization CRI profiles are stable at all altitudes; the averaged values are presented in Table 2 as column-integrated values. The $\alpha^{ext}$ profile for 355/532 nm is close to 0 at all altitudes.

For the comparison with NMMB/BSC-Dust model, the mass concentration profile needed to be obtained. In case of a high predominance of coarse aerosol mode, particle density was taken to be equal to 2.6 $g/cm^3$. The mass concentration profiles obtained by GARRLiC and LIRIC algorithms are close to 1170 $\mu g/m^3$ at 1.5 km, which is slightly higher in comparison with the NMMB/BSC-Dust model result (900 $\mu g/m^3$). Similarly, the night-time mass concentration is close to 500 $\mu g/m^3$,

whereas the modelled value is close to 1700 $\mu g/m^3$ at the same altitude of 1.5 km.

### 4.3    Analysis of a heavy dust event in Dakar on 10 April 2015

The third and the last dust event considered in our study was observed on 10 April 2015 over Dakar (11 days later). Three time ranges were selected for the analysis; the first two during day-time (15:00–19:00 for Raman and 16:01–16:19 for GARRLiC and LIRIC) and the third during night-

time (21:00–04:00 of 11 April 2015 for Regularization). The atmospheric conditions were stable, but the height of the aerosol layer containing almost all the aerosols increased from 3 to 4.5 km from the day- to night-time measurements. The day-time event was characterized by a high aerosol load (AOD 440 nm $\approx 1.53 \pm 0.04$; $\alpha \approx 0.02 \pm 0.01$ for 440/870 nm). Unfortunately, there were no lunar-photometer measurements because of the lunar phase. However, AOD derived by the integration of

the $\sigma$ profile obtained by the Raman method at 532 nm wavelength is equal to 0.83. It should be noted that such an estimation of AOD does not include aerosols located in the overlap zone of LIDAR. The NMMB/BSC-Dust model forecasted a dust event over the Dakar site with AOD values ranging from 0.8 to 1.6 at 550 nm for both day- and night-time (Fig. 13). Unfortunately, no measurements could be obtained from the wind LIDAR. However, a sea breeze was observed at the ground level during

day-time measurements. Back-trajectory analysis showed that during the day-time event, the sources of air-masses that came to the observational site changed from north (coast of Mauritania) at 0.5 km to east (north-west of Mali) at 2.5 km (Fig. 14). And, during the night, air-masses were coming from the east-north direction (Sahara region) at all altitudes. The GARRLiC and Raman + Regularization aerosol retrievals for the day- and night-time measurements are presented in Table 3.

GARRLiC single-mode inversion was used because of the huge predominance of coarse mode particles. As in the previous event, day-time $r_{eff}$ is high and equal to 2.0 $\mu m$, and the value decreases to 0.9 $\mu m$ at night. Th day-time column integrated RRI is close to $1.59 \pm 0.02$ and stays rather stable



**Table 3.** Aerosol properties during the dust event over the Dakar site on 10 April 2015. The LR values marked by ** were retrieved by the LIRIC algorithm. Only the values given for all the wavelengths refer to the column-integrated property. Day: AOD 440 nm $\approx 1.53 \pm 0.04$; $\alpha \approx 0.02 \pm 0.01$. Night: AOD 532 nm $\approx 0.83$; $\alpha \approx 0$ by Raman.

| $\lambda$ [nm] | GARRLiC | | | | | AERO-NET | Raman (Day) | Raman + Regularization (Night) | | | |
|---|---|---|---|---|---|---|---|---|---|---|---|
| | $r_{eff}$ [$\mu m$] | Sph % | RRI | IRI | LR [sr] | LR [sr] | LR [sr] | $r_{eff}$ [$\mu m$] | RRI | IRI | LR [sr] |
| 355 | | | 1.60 | 0.004 | 20 | 70** | ∼25 | | | | ∼59 |
| 440 | | | 1.60 | 0.003 | 17 | 62 | | | | | |
| 532 | | | 1.60 | 0.003 | 14 | 49** | ∼23 | | | | ∼50 |
| 675 | 2.0 | 57% | 1.60 | 0.002 | 13 | 39 | | 0.9 | 1.54 | 0.008 | |
| 870 | | | 1.59 | 0.002 | 12 | 32 | | | | | |
| 1020 | | | 1.58 | 0.002 | 13 | 31 | | | | | |
| 1064 | | | 1.58 | 0.002 | 13 | 30** | | | | | |

at night-time ($1.54 \pm 0.06$). The IRI slightly decreases during the day-time from 0.004 to 0.002 in the UV–near-infrared range, and is close to $0.008 \pm 0.004$ at the night-time. The GARRLiC SSA
increases from 0.85 to 0.95 in UV–near-infrared range. The maximum of SD is shifted to higher radii (Fig. 15). However, in general, RRI, IRI, SSA, and SD retrieved by GARRLiC are quite comparable to AERONET values. However, $r_{eff}$ and particle sphericity differ. AERONET $r_{eff}$ is equal to 1.6 $\mu m$ and sphericity ∼0%, while the GARRLiC algorithm retrieved 2.0 $\mu m$ for $r_{eff}$ and 57% of sphericity particles.
Volume concentration profiles are presented in Fig. 16. Because of different AOD values and altitudes of the boundary layer, day- and night-time V are not comparable. The LIRIC and GAR-RLiC day-time V are different, especially below 2 km, it can be explained by LIRIC usage of both 532 nm parallel and perpendicular signals whereas GARRLiC used total backscattered signal only. However, obtained uncertainties are high and data are overlapped. Extinction profiles (Fig. 17) at
all wavelengths are close to each other in respect to the retrieval algorithm. Differences between LR retrieved by the algorithms are very high, but nevertheless, $\sigma$ profiles of different algorithms do not have such high differences. The Raman LR values at 532 nm increase from ∼23 to ∼50 sr over the day- to night-time measurement timeframe, and LR at 355 nm also increases from ∼25 to ∼59 sr (Fig. 18). Such an increase in LR from day- to night-time measurements can be explained by the
contribution of marine particles during the day-time (i.e. sea breeze effect). Particle depolarization is lower during day-time (29%) than during night-time (32%). The Raman $\alpha_{ext}$ profile at 355/532 nm is close to 0 and does not change with altitude (this is not shown in the figures). The profile





of the effective radius retrieved with Regularization decreases from 1.2 $\mu m$ to 0.6 $\mu m$ at the altitude range of 1–4.5 km. Regularization RRI and IRI profiles remain stable through all altitudes, and

column-integrated values, which are presented in Table 3, have been taken as averaged values.

The volume concentration profile at 2 km is much higher during the day-time measurements than the one obtained at night. For comparison with NMMB/BSC-Dust model results, the mass concentration profile was obtained. Similar to the previous event, the particle density was taken to be equal to 2.6 $g/cm^3$ according to the NMMB/BSC-Dust model. The obtained mass concentration

profiles at 2 km are close to 1225±400 and 1020±90 $\mu g/m^3$ for GARRLiC and LIRIC, respectively. This is at least two times higher in comparison with the value produced by the NMMB/BSC-Dust model (∼550 $\mu g/m^3$). Such high differences can be explained in terms of particle coating processes, which probably had taken place because of sea breeze and high humidity (Binietoglou et al., 2015; Veselovskii et al., 2010). Similarly, the calculated night-time mass concentration (310±60 $\mu g/m^3$)

shows good agreement with the NMMB/BSC-Dust model (∼300 $\mu g/m^3$) at 2 km.

Raman and GARRLiC day-time LR indicate very likely the measurements of marine particles, but at the same time, the depolarization ratio indicates at mineral dust. The GARRLiC results are more consistent with mineral dust, but at the same time, the retrieved sphericity (57%) is too high for dust and LR values at all wavelengths are too low. Probably, a dust coating process played a role

in this event.

Such a complex event, which includes several types of particles (marine, mineral dust, and coated dust) with similar radii, can cause difficulties in retrieving, interpreting, and comparing the results. The GARRLiC and LIRIC height resolved aerosol properties are incompatible with the NMMB/BSC-Dust V and Raman $\sigma$ profiles. That is why, to avoid inconsistencies between the results of different

methods, GARRLiC should be only implemented in cases (*i*) where a single aerosol type is present or (*ii*) when the investigated aerosols can be separated into two different types of fine and coarse modes.

## 5   Conclusions

As was mentioned above, the main objective of this article is to compare aerosol properties retrieved

by different algorithms. This helps to know, to what extend these algorithms can be used in a complementary way for long-term day–night aerosol observations and data processing.

Three dust events were selected from LILAS measurements. The first event over Lille on 30 March 2014 was characterized by transported mineral dust particles from the Saharan region. Three different layers of aerosols were observed: (*i*) continental clean aerosols up to 2.5 km, (*ii*) mixtures

of continental and dust aerosols in the altitude range of 2.5 km to 6 km, and (*iii*) cirrus clouds with a negligible AOD impact at heights of 11 to 12 km.




The second and third events over Dakar were characterized by a layer consisting of a dust–marine aerosol mixture during the day-time and only dust particles during the night-time. In both cases, AOD values decrease over the day- to night-time measurement timeframe, and therefore, it was not possible to compare the day- and night-time $\sigma$. GARRLiC, LIRIC, and Raman day-time $\sigma$ profiles are in agreement on 29 March. However, $\sigma$ profiles retrieved by the same algorithms on 10 April differ. The latter was a more complex event with different types of particles in one aerosol mode. Development, such as introducing depolarization profile into the GARRLiC algorithm, should enhance the algorithm and make it possible to distinguish aerosols with different shapes inside one mode. In both dust cases, $r_{eff}$ were found to be higher during day-time in comparison with the night-time cases, probably due to the presence of marine particles. Raman LR increased over the day- to night-time measurement timeframe, and this also could have been caused by the presence of less marine particles at night. However, GARRLiC LR values were always lower than the ones obtained by LIRIC and Raman. Also, GARRLiC sphericity was always higher than the one obtained by AERONET. The latter two features can be caused by additional LIDAR information presented into the algorithm, which corrected the modelling of phase function at the $180^o$ direction (Dubovik et al., 2006). Hence, GARRLiC inversion without any additional data was performed to check the influence of adding LIDAR information, and the retrieved aerosol properties had similar values to AERONET products. Therefore, we assume that the results of LR, sphericity, and other properties retrieved by GARRLiC with the LIDAR data are more accurate than those derived without such information. Nevertheless, an accurate modelling of light scattering by non-spherical particles remains to be one of the major difficulties in remote sensing of tropospheric aerosols. The presence of marine particles should decrease RRI values during the day, but day-time RRI values were higher in comparison with the night-time ones. However, day-time IRI values were lower in comparison with the ones obtained at night. This agrees with presence of marine particles, which absorb less than dust particles. Nevertheless, depolarization ratios were always indicative of dust particles. More events should be analysed in order to distinguish the inconsistencies between the algorithms. The second phase of the SHADOW2 campaign is taking place in December–January 2016.

In both cases, there were marine particles present during the day-time measurements. In future studies, it will be interesting to select morning measurements excluding see-breeze and marine particles, since humid environments can change the optical properties of aerosols by coating. Such an effect can be important for the exact quantitative characterization of dust (e.g. volume concentration). However, it does not completely prevent studying the mix of dust. The coating effect can have a large impact in case of synergetic sun/sky-photometer and LIDAR data retrieval. The overestimation of the retrieved aerosol volume concentration by GARRLiC and LIRIC during the third event was likely caused by this. Consequently, there were inconsistencies between the retrieved and modelled mass concentrations. GARRLiC development (for instance, by incorporating the Raman technique and/or depolarization profile into the code) will make it possible to distinguish vertically resolved





aerosol optical properties more accurately, i.e. improved extinction and volume concentration pro-
475 files. After such improvements, similar studies should be carried out, and again, the algorithm results
should be compared to determine if they are able to complement each other for long-term day–night
measurements.

*Acknowledgements.* The research leading to these results has received funding from European Union's Hori-
zon 2020 research and innovation programme under grant agreement No 654109 within ACTRIS project.
480 The authors acknowledge Diaollo Aboubacry and Thierno Ndiaye for IRD (Institute pour la Recherche et le
Développement) at Mbour, Dakar, Senegal and CaPPA project for the its support.





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

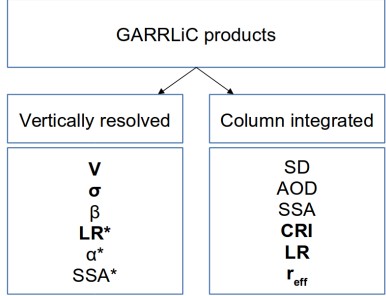

**Figure 1.** GARRLiC products derived by using single (unmarked) or double (both unmarked and marked *) mode inversion. The latter can be applied only to multi-wavelength LIDAR data. Common properties, which are compared in this work, retrieved using GARRLiC, LIRIC, and Raman + Regularization are indicated by bold font.

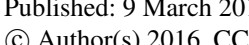



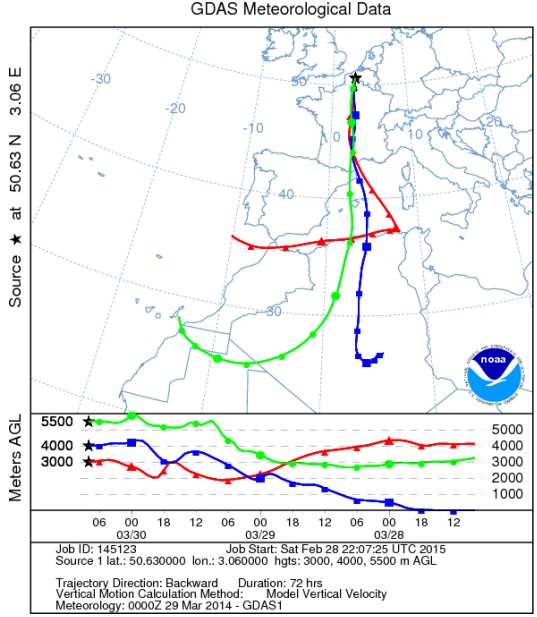

**Figure 2.** Backward trajectories of air-masses observed over Lille during the morning of 30 March 2014.

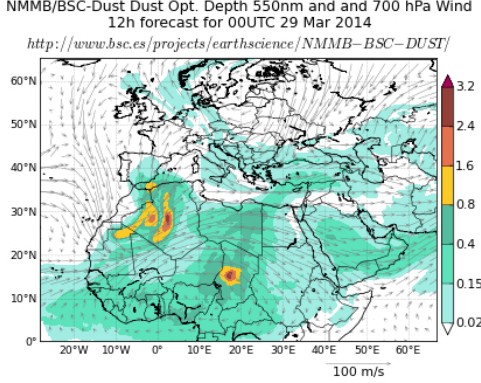

**Figure 3.** Dust event over Algeria on 29 March 2014.




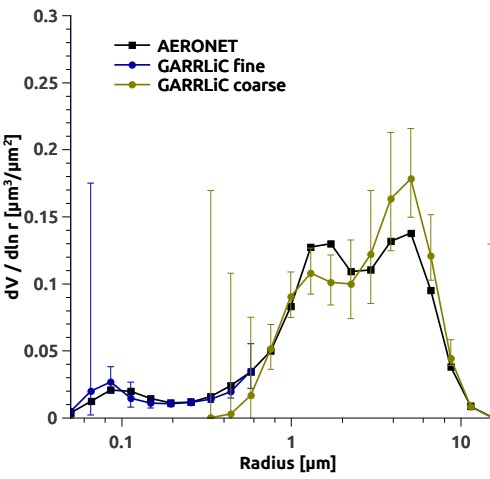

**Figure 4.** Volume size distribution retrieved by GARRLiC (blue and dark yellow correspond to the fine and coarse modes, respectively) and AERONET (black) on 30 March 2014 (7:40 UTC) in Lille (AOD 440nm $\approx 0.52; \alpha \approx 0.27$ for 440/870 nm).

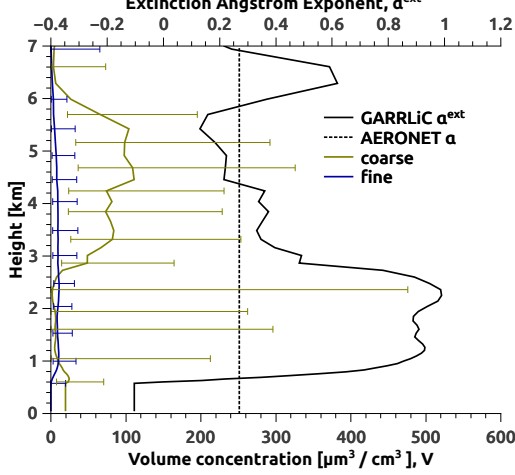

**Figure 5.** Volume concentration (V, blue and dark yellow correspond to the fine and coarse modes, respectively) and the extinction Ångström exponent (black, $\alpha^{ext}$) retrieved by GARLLiC; the AERONET Ångström exponent value is shown by the dashed black line. Data were carried out on 30 March 2014 over Lille.





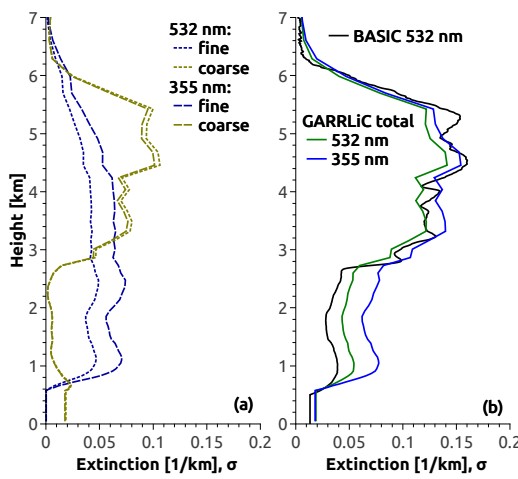

**Figure 6.** Aerosol extinction profiles retrieved by the GARRLiC and the BASIC algorithms for dust event over Lille on 30 March 2014. Left panel (a) presents the aerosol mode contribution of $\sigma$ at each wavelength, which was retrieved by using GARRLiC; right panel (b) presents the total GARRLiC $\sigma$ at wavelengths of 355 and 532 nm and BASIC $\sigma$ at a wavelength 532 nm.

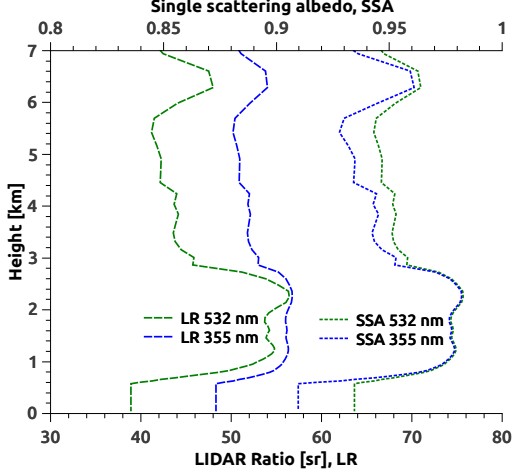

**Figure 7.** GARRLiC LR and SSA for an event over Lille on 30 March 2014.





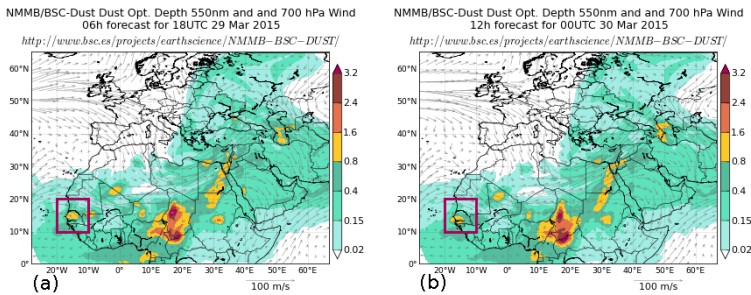

**Figure 8.** NMMB/BSC-Dust model over Africa and Europe on 29 March 2015. AOD values forecasted by the model ranged from 0.8 to 1.6 at 550 nm. (a) 18:00 UTC, 29 March; (b) 00:00 UTC, 30 March.

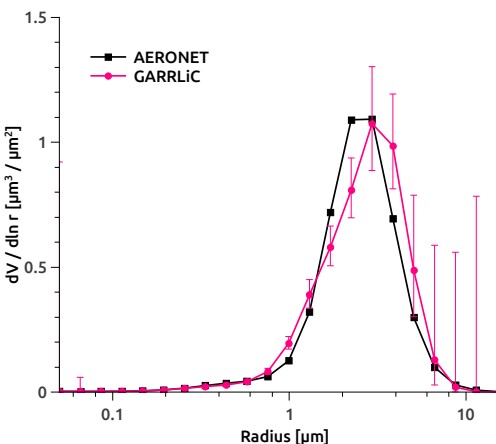

**Figure 9.** GARRLiC (pink) and AERONET (black) SD on 29 March 2015 (16:49 UTC) over the Dakar site (AOD 440 nm $\approx 1.35 \pm 0.20$; $\alpha \approx -0.04 \pm 0.01$).





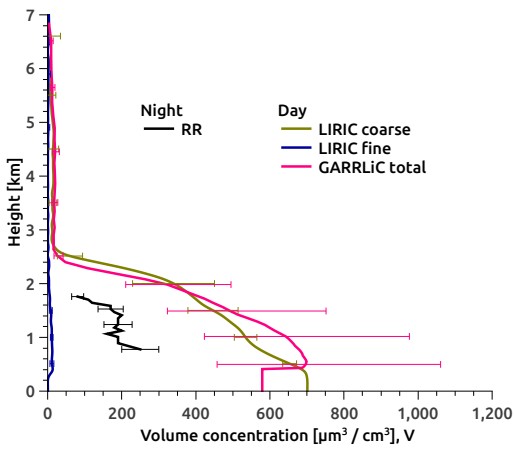

**Figure 10.** Volume concentration profiles for an event over the Dakar site on 29 March 2015. The abbreviation RR corresponds to V retrieved by using Raman + Regularization algorithms.

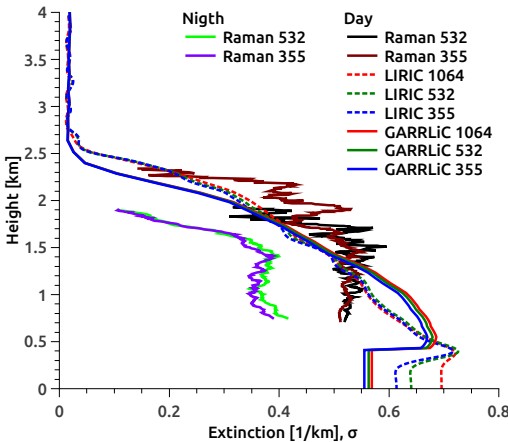

**Figure 11.** Raman, LIRIC, and GARRLiC extinction profiles during the day (AOD 440 nm $\approx 1.35 \pm 0.20$; $\alpha \approx -0.04 \pm 0.01$) and night (AOD 440 nm $\approx 0.83 \pm 0.03$; $\alpha \approx 0.08 \pm 0.02$) on 29 March 2015.





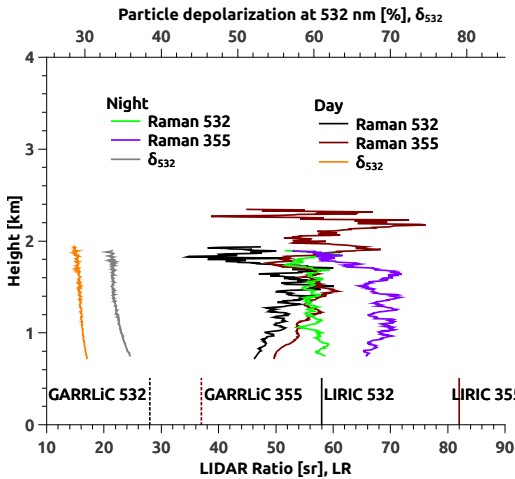

**Figure 12.** LIDAR ratio and depolarization ratio during day- and night-time measurements for an event on 29 March 2015 over the Dakar site. GARRLiC and LIRIC LR column-integrated values are shown at the beginning of the profiles.

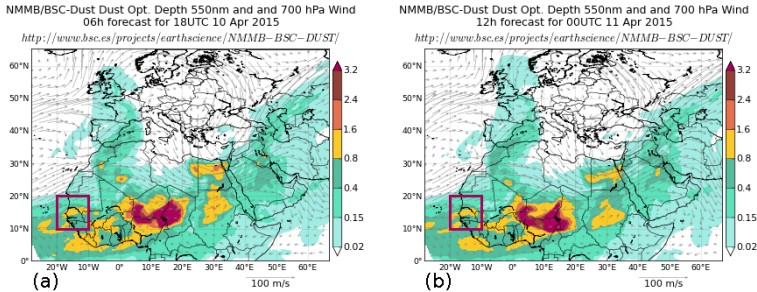

**Figure 13.** NMMB/BSC-Dust model results over Africa and Europe on 10 April 2015. AOD values forecasted by the model ranged from 0.8 to 1.6 at 550 nm. (a) 18:00 UTC, 10 April; (b) 00:00 UTC, 11 April.



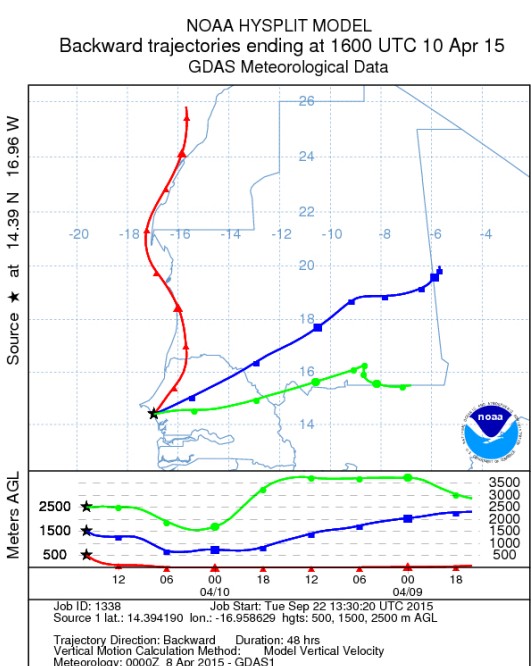

**Figure 14.** Backward trajectories of air-masses for an event over the Dakar site on 10 April 2015.





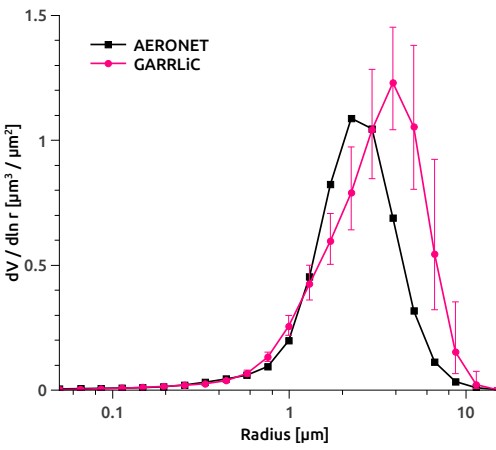

**Figure 15.** GARRLiC (pink) and AERONET (black) SD for 10 April 2015 (16:11 UTC) over the Dakar site
(AOD 440 nm ≈ $1.53 \pm 0.04$; $\alpha \approx 0.02 \pm 0.01$).

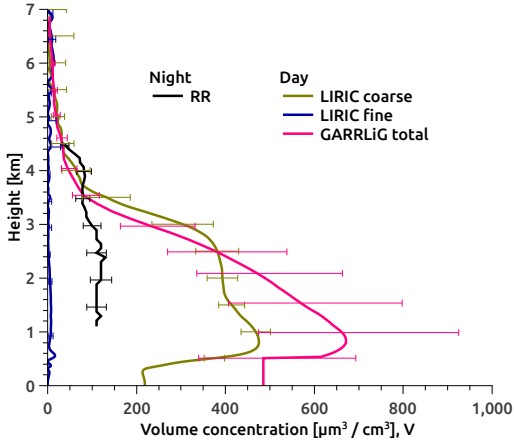

**Figure 16.** Volume concentration profiles for an event over the Dakar site on 10 April 2015. The abbreviation
RR corresponds to V retrieved by using Raman + Regularization algorithms.





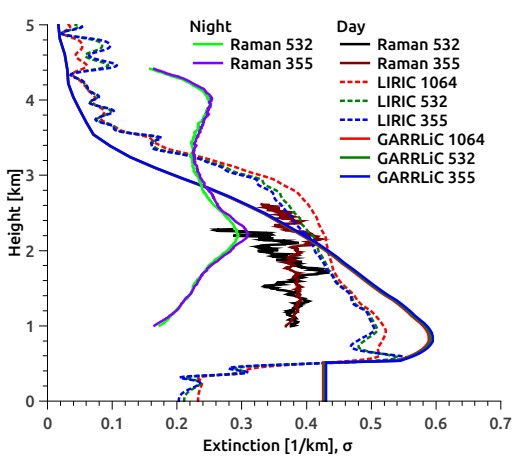

**Figure 17.** Raman, LIRIC, and GARRLiC extinction profiles during the day (AOD 440 nm $\approx 1.53 \pm 0.04$; $\alpha \approx 0.02 \pm 0.01$) and night (AOD 532 nm $\approx 0.83$; $\alpha \approx 0$ by Raman) on 10 April 2015.

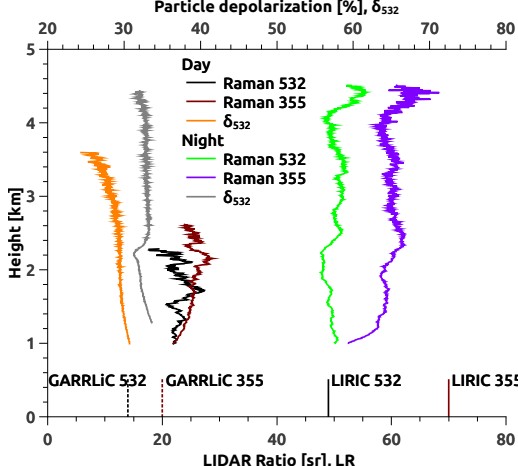

**Figure 18.** LIDAR ratio and depolarization ratio during day- and night-time measurements for an event on 10 April, 2015 over the Dakar site. GARRLiC and LIRIC LR column integrated values are shown at the beginning of the profiles.