# Peer review of "Comparison of aerosol properties retrieved using GARRLiC, LIRIC, and Raman algorithms applied to multi-wavelength LIDAR and sun/sky-photometer data"

_Atmospheric Measurement Techniques, 2016_

## Referee Comment (RC1) · Anonymous Referee #1 · 31 Mar 2016

Depending on the LIDAR characteristics, different techniques can be used for obtaining optical and microphysical properties of aerosols and is difficult to decide in some particular cases which one of these techniques is the best and provide the best output. This paper is focused on detailed analysis and comparison of some aerosol optical properties and microphysical properties retrieved using The Raman technique and inversion with regularization, LIRIC (LIdar-Radiometer Inversion Code) and GARRLiC (Generalized Aerosol Retrieval from Radiometer and LIDAR Combined data). There are some inconsistencies in the retrieved LIDAR ratios, but aerosol properties derived from lidar measurements were generally in agreement with the AERONET (AErosol RObotic

[Figure]

NETwork). Three dust events were presented in details: over Lille on 30 March 2014 transported mineral dust, analysis of a heavy dust event in Dakar on 10 April 2015 and on 29 March 2014 with complex aerosol load. More papers like this should be published so we could get a better view of the possibilities and challenges in using these different algorithms.
* * *

---

## Referee Comment (RC2) · Anonymous Referee #3 · 4 Apr 2016

The manuscript by Bovchaliuk et al compares results obtained for dust-dominated aerosols from different algorithms in three case studies. Overall the manuscript is in a good shape, but discussion is improvable. The manuscript is of interest for AMT readers and thus should be published after some clarifications and revisions.

List of comments:

Line 15: "It" –> "In"

Line 19: It is, in general, not correct that in situ measurements involves direct measurements. There may be some types of in-situ measurements that directly measure the

quantity of interest, but most need data inversion. For example, measured scattered light gets inverted to particle size.

Line 22f.: There are also passive remote sensing instruments measuring terrestical radiation (infrared, microwave).

Line 33: "Spatial" could be removed.

Line 60: You should already mention here the name of the lidar system "LILAS".

Line 66: "a multi-wavelength LIDAR" could be replaced by "LILAS".

Line 100: It is unclear what BASIC means here. My reader would first think that this is the name of the programming language used. From reading on it becomes clear that this is the name of an algorithm. To avoid confusion, I suggest to add in parentheses the written-out name of this algorithm.

Line 110: "scattered sunlight by atmospheric molecules": Not only molecules but also aerosols and clouds could be a problem. I suggest "sunlight scattered into the field of view of the lidar".

Section 3: Please mention that the Regularization algorithm assumes spherical particles (though it is mentioned in the reference, it is an important aspect that should be mentioned also in your manuscript).

Line 195: "origin of aerosols from mineral dust": Shouldn't origin be a location, like the Sahara?

Line 211: I'm surprised about the low uncertainty. If applying spherical particles to mineral dust the error can be significantly larger than 20%. This should be mentioned and considered in the discussion of your results.

Line 213: What means "mineral dust with inclusions of marine particles"? I guess you mean "mineral dust sometimes mixed with marine aerosol particles".

Line 228f.: It is unclear what the difference between "dust intensity" and "dust load" is.

Line 258: It should be discussed why the BASIC LR value at 355 nm is so much lower than all others.

Line 261f.: "the second one with larger radii refers to the particles of thin cirrus clouds": This is speculation and contradicts the statement in line 236. In addition, in most cases cirrus clouds, particles would be larger than the size at your second maximum.

Line 310: "and decreases to 1.1 microns at night". But using a different method, so the "true" effective radius may have not changed.

Line 311f.: Lower RRI could be also effect of method.

Line 314-323: But the depolarization ratio shows that there was at most a very little contribution of marine aerosol during day-time. The depolarization ratio is ∼29% compared to ∼34% at night, but given the low lidar ratio of marine aerosol (15-20sr) compared to desert dust (45-60sr) the contribution of marine aerosol to extinction must have been <10% during day-time. Thus your discussion is not convincing to me.

Line 335f.: "Unfortunately, a comparison between the day- and night-time V values was not possible ... " But this is shown in Fig. 10. Please clarify.

Line 338: Again the 20% uncertainty seems unrealistically low.

Line 342: "were averaged into 60 points" is unclear.

Line 343: "The day-time Raman LR values decrease with altitude": Fig. 12 looks different.

Line 344: What means "slightly correct"?

Line 371: What is the "overlap zone"? Zone of full overlap or zone of incomplete overlap?

Line 381f.: Again, the decrease from day to night could be a method effect, which

should be considered better in your discussion.

Line 382: "Th" –> "The"

Section 4.3: The measured combination of lidar ratio and depolarization ratio is quite unique. The depolarization ratio indicates Saharan dust whereas the lidar ratio indicates dominance of sea salt. "Probably, a dust coating process played a role in this event." (line 419) seems very speculative to me. What I'm missing in this section is a discussion of realistic measurement uncertainties, in particular of the LR, and also of the possibility that sea salt got into a dry state (r.h. $\sim < 50\%$). In addition, what confuses me is the high RRI from Garrlic which indicates that the contribution of marine aerosol was not significant (since marine aerosol has a much lower RRI). This should be considered in your discussion.

Line 393: Would it be possible or have you tried to use parallel and perpendicular signals also with GARRLIC?

Line 434: "Continental clean" seems wrong.

Line 442: "in one aerosol mode" might be replaced by "in the same size range".

---

## Referee Comment (RC3) · Anonymous Referee #2 · 5 Apr 2016

Lines 135-136: "Such differences. . . significantly"

Has this been shown? If yes, please provide the reference. To my knowledge, previous studies showed that GARRLiC and LIRIC agree well. Since you also do not show "significant" differences in the cases you present here, please rephrase this statement.

Line 277: ". . .typical for urban-industrial particles"

Please provide reference.

Line 390: "Volume concentration profiles are presented in Fig. 16"

[Figure]

Plot the total LIRIC concentration (fine+coarse) to compare with the one-mode (total) GARRLiC concentration.

Lines 450-457: "The latter two features... tropospheric aerosols."

The model used for the non-spherical particle calculations has not been proven to work for backscattering in any case. There are many publications that show that it does not work always accurately (e.g. Wiegner, M., J. Gasteiger, K. Kandler, B. Weinzierl, K. Rasp, M. Esselborn, V. Freudenthaler, B. Heese, C. Toledano, M. Tesche, and D. Althausen (2009), Numerical simulations of optical properties of Saharan dust aerosols with emphasis on lidar applications, Tellus B, 61, 180-194, doi: 10.1111/j.1600-0889.2008.00381.x.)

So, the fact that GARRLiC includes the lidar measurements in the retrieval does not make it more accurate, if the model it uses cannot reproduce the 180o measurements well, for all cases. The differences between the algorithms cannot be explained following the reasoning in the text. You should state the problems with the non-spherical particle modelling, providing relevant literature and conclude that based on this, the results are not conclusive and that more work needs to be done with respect to their validation.

---

## Short Comment (SC1) · 3 May 2016

I would like to make the following comments, related to the consistency within the paper.

If I understood correctly, the Aeronet retrievals are given for comparison purpose only (in addition to the three methods discussed).

case 1) LIRIC and Raman retrievals are missing and basically do not contribute to the inter-comparison among the three methods

cases 2) and 3) column integrated variables (table 2): please provide Aeronet CRI and LIRIC CRI

vertically resolved variables: please use fine and coarse modes for GARRLIC to be consistent with LIRIC retrievals

For consistency, the same input (photometer + lidar) should be used in both LIRIC and GARRLIC. Apologises if I missed something!

---

## Author Comment (AC1) · 28 May 2016

Thank you for comments! The responses structured as it suggested by editorial: (1) comments from Referees, (2) author's response, (3) author's changes in manuscript.

(1) Line 15: "It" –> "In" (2) Thank you, I have change it. (3) "In future studies..."

(1) Line 19: It is, in general, not correct that in situ measurements involves direct measurements. There may be some types of in-situ measurements that directly measure the quantity of interest, but most need data inversion. For example, measured scattered light gets inverted to particle size. (2) I have deleted the word direct. Hence,

it includes now both direct measurements of particles and measurements that needs data inversion. (3) "The former involves measurements of particles using instruments at the survey points."

(1) Line 22f.: There are also passive remote sensing instruments measuring terrestical radiation (infrared, microwave). (2, 3) Thank you. The sentence changed to "Instruments belonging to the passive category measure the modified solar radiation after interactions with particles and terrestrial radiation."

(1) Line 33: "Spatial" could be removed. (2) Done (3) "The main advantages of LIDAR measurements include high vertical resolution and applicability during night-time and in cloudy environments."

(1) Line 60: You should already mention here the name of the lidar system "LILAS". (2) Name of LIDAR instrument LILAS added to the sentence. (3) "The LIDAR system LILAS used in this work..."

(1) Line 66: "a multi-wavelength LIDAR" could be replaced by "LILAS". (2) Done (3) "LILAS was moved to M'Bour city (Dakar site) in Senegal..."

(1) Line 100: It is unclear what BASIC means here. My reader would first think that this is the name of the programming language used. From reading on it becomes clear that this is the name of an algorithm. To avoid confusion, I suggest to add in parentheses the written-out name of this algorithm. (2) I have rewritten the sentence in hope it will help to avoid the confusion. (3) "The algorithm called BASIC based on the Klett method..."

(1) Line 110: "scattered sunlight by atmospheric molecules": Not only molecules but also aerosols and clouds could be a problem. I suggest "sunlight scattered into the field of view of the lidar". (2) Yes, it is true. It is very critical to understand, thank you for pointing me out! (3) "It is mostly used at night-time when the signal to noise ratio is the highest, owing to the absence of sunlight scattered into the field of view of the lidar."

(1) Section 3: Please mention that the Regularization algorithm assumes spherical particles (though it is mentioned in the reference, it is an important aspect that should be mentioned also in your manuscript). (2) The model of polydisperse, randomly oriented spheroids have been used for article's cases. I apologize that it have not been mentioned in the article. Also, I forgot to add reference for the article (I. Veselovsky, et al., Application of randomly oriented spheroids for retrieval of dust particle parameters from multiwavelength lidar measurements, 2010). I have added the reference.

(1) Line 195: "origin of aerosols from mineral dust": Shouldn't origin be a location, like the Sahara? (2) Yes, it should be. Sentence have been changed: (3) "...confirmed the origin of mineral dust from Sahara and showed the source locations"

(1) Line 211: I'm surprised about the low uncertainty. If applying spherical particles to mineral dust the error can be significantly larger than 20%. This should be mentioned and considered in the discussion of your results. (2) Sorry for such confusion, we have applied model of polydisperse, randomly oriented spheroids.

(1) Line 213: What means "mineral dust with inclusions of marine particles"? I guess you mean "mineral dust sometimes mixed with marine aerosol particles". (2) Have been changed. (3) "As this work mainly deals with mineral dust sometimes mixed with marine aerosol particles..."

(1) Line 228f.: It is unclear what the difference between "dust intensity" and "dust load" is. (2) In comparison with dust load over Darak site this event over Lille characterized as moderate, but for Lille site such AOD is high. I have deleted "as moderate in terms of dust intensity, but", hence it is clear now: (3) "The dust event detected over Lille on 30 March 2014 was characterized as heavy for Lille site in terms of the aerosol load..."

(1) Line 258: It should be discussed why the BASIC LR value at 355 nm is so much lower than all others. (2) Thank you for pointing me out! I apologize for such inconvenience, I have uncommented and transfer sentences (269f-273) to line 245. It is very important part, here it is: (3) "Is should be emphasize, depolarization calibration of 355

nm have not been done for the event. Hence, only parallel channel considered in GAR-RLiC and BASIC algorithms. Also, it should be noted that the BASIC algorithm works based on mono-wavelength LIDAR signals. In case of multi-wavelength LIDAR measurements, BASIC retrieves aerosol properties individually at each wavelength. Hence, its results at 355 nm should not be considered as satisfactory. Nevertheless, BASIC LR is presented in Table 1 just for acknowledgment. In case of GARRLiC retrieval, absent of 355 perpendicular channel in total signal had been probably neglected by simultaneous retrieval of all LIDAR and radiometric data."

(1) Line 261f.: "the second one with larger radii refers to the particles of thin cirrus clouds": This is speculation and contradicts the statement in line 236. In addition, in most cases cirrus clouds, particles would be larger than the size at your second maximum. (2) Probably I do not understand the question, but why it contradicts to criteria of cloud detection? If the cirrus cloud is homogeneous all around the whole sky than it is hard to detect cloud by left\right sky radiance criteria. In most cases particles of cirrus clouds are larger, but a possibility that second maxima refers to particles of cirrus clouds should not be rejected here. (3) I have changed this sentence to: "The first one with lower radii likely indicates dust particles, and the second one with larger radii also indicates dust particles of can refer to the particles of thin cirrus clouds. ".

(1) Line 310: "and decreases to 1.1 microns at night". But using a different method, so the "true" effective radius may have not changed. Line 311f.: Lower RRI could be also effect of method. Line 314-323: But the depolarization ratio shows that there was at most a very little contribution of marine aerosol during day-time. The depolarization ratio is âĹij29% compared to âĹij34% at night, but given the low lidar ratio of marine aerosol (15-20sr) compared to desert dust (45-60sr) the contribution of marine aerosol to extinction must have been <10% during day-time. Thus your discussion is not convincing to me. (2, 3) I have changed last sentence to: "Such a behavior of retrieved aerosol properties could be explained by the influence of marine aerosols during day-time and/or using of different methods."

(1) Line 335f.: "Unfortunately, a comparison between the day- and night-time V values was not possible ... " But this is shown in Fig. 10. Please clarify. (2) The main goal of article is to compare aerosol properties obtained using different algorithms. AOD have been changed significantly and that is why it is impossible to conclude if the algorithms are in agreement and can complement each other. (3) I have modified this sentence to: "Unfortunately, a comparison in terms of obtaining close V values using GARRLiC, LIRIC and Regularization algorithms between the day- and night-time are not possible because of a significant decrease in the AOD values."

(1) Line 342: "were averaged into 60 points" is unclear. (2) It is time-consuming to retrieve aerosol properties for all points of LIDAR signal for GARRLiC algorithm, that is why signals reduced by averaging into 60 point in preparation phase for GARRLiC algorithm. (3) I have change the sentence to: "...LIDAR signals were reduced by averaging into 60 points during the data preparation phase."

(1) Line 343: "The day-time Raman LR values decrease with altitude": Fig. 12 looks different. (2) Yes, thank you. (3) "The day-time Raman LR values increase with altitude"

(1) Line 344: What means "slightly correct"? (2) In case of column-integrated (GARRLiC, LIRIC in this case) LR retrieved extinction profile totally repeat backscatter profile in regard to LR coefficient. In case of vertically resolved LR extinction profile slightly corrects in accordance with LR coefficient, in this case it increases with altitudes. Probably, word slightly not correct here, I have delete it. (3) "...increase with altitude and, therefore, correct $\sigma$ profiles;"

(1) Line 371: What is the "overlap zone"? Zone of full overlap or zone of incomplete overlap? (2) In this case incomplete overlap. (3) Sentence have been changed to: "It should be noted that such an estimation of AOD does not include aerosols located in incomplete overlap zone of LIDAR."

(1) Again, the decrease from day to night could be a method effect, which should be considered better in your discussion. (2, 3) I have change the sentence in conclusion

at line 445: "In both dust cases, $r_{eff}$ were found to be higher during day-time in comparison with the night-time cases, probably due to the presence of marine particles or caused by using different methods."

(1) Line 382: "Th" –> "The" (2, 3) Done, thank you.

(1) Section 4.3: The measured combination of lidar ratio and depolarization ratio is quite unique. The depolarization ratio indicates Saharan dust whereas the lidar ratio indicates dominance of sea salt. "Probably, a dust coating process played a role in this event." (line 419) seems very speculative to me. What I'm missing in this section is a discussion of realistic measurement uncertainties, in particular of the LR, and also of the possibility that sea salt got into a dry state (r.h. âĹij< 50%). In addition, what confuses me is the high RRI from Garrlic which indicates that the contribution of marine aerosol was not significant (since marine aerosol has a much lower RRI). This should be considered in your discussion. (2) Due to weak Raman signal at 408 nm it is impossible to calculate RH profile using day-time LIDAR measurements. There are radiosonde measurements, but the nearest one was at 12UTC at Dakar, before the sea breeze have started (it has low RH close to 20-30% in altitude range up to 4km). Hence, there are no trustworthiness data about RH at the 15:00-19:00 time-range. In my opinion, the critical issue here is time-ranging. Sea-breeze started at 16:00. AERONET and LILAS data discussed in article close to ∼16:11, which means that there were small number of marine aerosols, hence they have small impact on GARRLiC retrieval. That is why GARRLiC RRI so high. Data for Raman algorithm have been taken from 15:00 to 19:00, so it includes part of measurements without sea breeze (15:00-16:00) and main part (16:00-19:00) of measurements within sea breeze. GARRLiC retrieved similar RRI values to AERONET ones. If you look at AERONET RRI values you can see that they are decreasing with time (at 16:11 values close to ∼1.59, at 16:50 ∼1.58, at 17:14 ∼1.57, and at 17:58 ∼1.55). This can indicate that more and more marine aerosol were coming during the event. But, in the same time SSA doesn't change (at all measurements it is increasing from 0.86 at 440 nm to 0.97

at 1024 nm). AERONET LR values, like SSA, indicate at dust particles. But, GARRLiC sphericity (57%) is too high for dust and LR values at all wavelengths are too low. All this have suggested to me that dust coating process were presented during sea breeze.

(1) Line 393: Would it be possible or have you tried to use parallel and perpendicular signals also with GARRLIC? (2) This part of algorithm is in developing mode. Hence, unfortunately, it is not yet possible to try parallel and perpendicular signals.

(1) Line 434: "Continental clean" seems wrong. (2, 3) I have changed to "continental".

(1) Line 442: "in one aerosol mode" might be replaced by "in the same size range". (2, 3) Done, thank you.

---

## Author Comment (AC2) · 31 May 2016

(1) Lines 135-136: "Such differences. . . significantly" Has this been shown? If yes, please provide the reference. To my knowledge, previous studies showed that GARRLiC and LIRIC agree well. Since you also do not show "significant" differences in the cases you present here, please rephrase this statement.

(2) Yes, it is try that previous studies shows agreement of results obtained using GARRLiC and LIRIC. But, both Dakar events shows significant differences in retrieved LR. From that and from algorithm's differences I have made a conclusion that retrieval results can significantly differs.

(3) I just delete the word "significantly". The sentence now is "Such differences in the algorithms can influence the results obtained by the two systems."

(1) Line 277: ". . .typical for urban-industrial particles" Please provide reference.

(2) Sorry, it is my misprint. Thank you very much for pointing me out!

(3) Changed to continental clean particles.

(1) Line 390: "Volume concentration profiles are presented in Fig. 16" Plot the total LIRIC concentration (fine+coarse) to compare with the one-mode (total) GARRLiC concentration.

(2) In Fig. 1 I plotted volume concentration profiles LIRIC (total, fine and coarse) and GARRLiC (total). I didn't plot errors to stay figure clear. Fine mode particles have small contribution in total volume concentration. Similar situation is for event on 29 March 2015.

(1) Lines 450-457: "The latter two features. . . tropospheric aerosols." The model used for the non-spherical particle calculations has not been proven to work for backscattering in any case. There are many publications that show that it does not work always accurately (e.g. Wiegner, M., J. Gasteiger, K. Kandler, B. Weinzierl, K. Rasp, M. Esselborn, V. Freudenthaler, B. Heese, C. Toledano, M. Tesche, and D. Althausen (2009), Numerical simulations of optical properties of Saharan dust aerosols with emphasis on lidar applications, Tellus B, 61, 180-194, doi: 10.1111/j.1600-0889.2008.00381.x.)

(2) Agree, I have rephrase sentence according to your comment.

(3) "The latter two features can be caused by additional LIDAR information presented into the algorithm, which corrected the modelling of phase function at the 180o (Dubovik, O., Sinyuk, A., Lapyonok, T., Holben, B. N., Mishchenko, M., Yang, P., Eck, T. F., Volten, H., Munoz,O., Veihelmann, B., et al.: Application of spheroid

models to account for aerosol particle nonsphericity in remote sensing of desert dust, Journal of Geophysical Research: Atmospheres (1984–2012), 111, 2006) direction or the model cannot reproduce well the measurements at the 180o (Müller, D., Veselovskii, I., Kolgotin, A., Tesche, M., Ansmann, A., and Dubovik, O.: Vertical profiles of pure dust and mixed smoke–dust plumes inferred from inversion of multiwavelength Raman/polarization lidar data and comparison to AERONET retrievals and in situ observations, Appl. Opt., 52, 3178–3202, doi:10.1364/AO.52.003178, http://ao.osa.org/abstract.cfm?URI=ao-52-14-3178, 2013; Wiegner, M., Gasteiger, J., Kandler, K., Weinzierl, B., Rasp, K., Esselborn, M., Freudenthaler, V., Heese, B., Toledano, C., Tesche, M., et al.: Numerical simulations of optical properties of Saharan dust aerosols with emphasis on lidar applications, Tellus B, 61, 180–194, 2009.) direction."
* * *
**Fig. 1.** LIRIC_fine-coare-total_GARRliC-total_volume_concentration

---

## Author Comment (AC3) · 1 Jun 2016

(1) Depending on the LIDAR characteristics, different techniques can be used for obtaining optical and microphysical properties of aerosols and is difficult to decide in some particular cases which one of these techniques is the best and provide the best output. This paper is focused on detailed analysis and comparison of some aerosol optical properties and microphysical properties retrieved using The Raman technique and inversion with regularization, LIRIC (LIdar-Radiometer Inversion Code) and GARRLiC (Generalized Aerosol Retrieval from Radiometer and LIDAR Combined data).

[Figure]

There are some inconsistencies in the retrieved LIDAR ratios, but aerosol properties derived from lidar measurements were generally in agreement with the AERONET (AErosol Robotic NETwork). Three dust events were presented in details: over Lille on 30 March 2014 transported mineral dust, analysis of a heavy dust event in Dakar on 10 April 2015 and on 29 March 2014 with complex aerosol load. More papers like this should be published so we could get a better view of the possibilities and challenges in using these different algorithms. (2) Thank you very much for such a good review!

---

## Author Comment (AC4) · 1 Jun 2016

(1) - short comment (2) - answer on short comment

(1) I would like to make the following comments, related to the consistency within the paper.

If I understood correctly, the Aeronet retrievals are given for comparison purpose only (in addition to the three methods discussed).

case 1) LIRIC and Raman retrievals are missing and basically do not contribute to the

inter-comparison among the three methods

(2) Yes, unfortunately there were no Raman channels at that time, that is why article doesn't include Raman retrievals. But, the event by itself is very interesting and I included it for GARRLiC/BASIC/AERONET comparison.

According to Chaikovsky et al. (Chaikovsky, A., Dubovik, O., Holben, B., Bril, A., Goloub, P., Tanré, D., Pappalardo, G., Wandinger, U., Chaikovskaya, L., Denisov, S., Grudo, Y., Lopatin, A., Karol, Y., Lapyonok, T., Amiridis, V., Ansmann, A., Apituley, A., Allados-Arboledas, L., Binietoglou, I., Boselli, A., D'Amico, G., Freudenthaler, V., Giles, D., Granados-Muñoz, M. J., Kokkalis, P., Nicolae, D., Oshchepkov, S., Papayannis, A., Perrone, M. R., Pietruczuk, A., Rocadenbosch, F., Sicard, M., Slutsker, I., Taliani, C., De Tomasi, F., Tsekeri, A., Wagner, J., and Wang, X.: Lidar-Radiometer Inversion Code (LIRIC) for the retrieval of vertical aerosol properties from combined lidar/radiometer data: development and distribution in EARLINET, Atmospheric Measurement Techniques Discussions, 8, 12 759–12 822, doi:10.5194/amtd-8-12759-2015, http://www.atmos-meas-tech-discuss.net/8/12759/2015/, 2015.) the results of the LIRIC inversion using only the data from 355 and 532 nm wavelengths were found to be rather unsatisfactory for the coarse mode, that is why LIRIC inversion had not been applied to this event too.

(1) cases 2) and 3) column integrated variables (table 2): please provide Aeronet CRI and LIRIC CRI

(2) According to Chaikovsky et al. (reference above), the LIRIC algorithm uses AERONET inversion products such as column volume concentration, volume specific backscatter, and extinction coefficients as a priori information. The specific products include backscatter ($\beta$), extinction ($\sigma$) and volume concentration (V) profiles, Angström exponent ($\alpha$) values, and LIDAR (LR) and depolarization ($\delta$) ratios. LIRIC doesn't retrieve CRI values. In attached supplement file you will find modified article's Tables 2 and 3 with AERONET CRI.

(1) vertically resolved variables: please use fine and coarse modes for GARRLIC to be consistent with LIRIC retrievals

(2) Because of high predominance of coarse particles, GARRLiC retrieval of fine mode has high errors and the results were found to be rather unsatisfactory. That is why both Dakar events were retrieved by using the configuration of single mode inversion. In attached supplement file you will find updated Figure 16 which have LIRIC (fine, coarse and total) and GARRLiC (total) volume concentrations. I didn't plot errors to stay figure clear. As you can see, fine mode particles have small contribution in total volume concentration. Similar situation is for event on 29 March 2015.

(1) For consistency, the same input (photometer + lidar) should be used in both LIRIC and GARRLIC. Apologises if I missed something!

(2) In both LIRIC and GARRLiC algorithms have been used same photometer and lidar data. Probably you were noticed that LIRIC and GARRLiC vertical profiles have different number of points. GARRLiC retrieval is time consuming, that is why during the GARRLiC data preparation phase lidar signals were reduced by averaging into 60 points. It reduce time of retrieval and errors in upper altitudes.

Please also note the supplement to this comment:
http://www.atmos-meas-tech-discuss.net/amt-2016-40/amt-2016-40-AC4-supplement.pdf

**Supplement:**

Manuscript prepared for Atmos. Meas. Tech.
with version 2015/11/06 7.99 Copernicus papers of the LaTeX class copernicus.cls.
Date: 1 June 2016

**Additional tables to article "Comparison of aerosol properties retrieved using GARRLiC, LIRIC, and Raman algorithms applied to multi-wavelength LIDAR and sun/sky-photometer data"**

V. Bovchaliuk[1]

[1]Laboratoire d'Optique Atmospherique, Lille1 University, Villeneuve d'Ascq, France

*Correspondence to:* Valentyn Bovchaliuk (bovchaliukv@gmail.com)

**Table 1.** = Tabel 2 from article. Aerosol properties during the dust event over the Dakar site on 29 March 2015. Here and further, the LR values marked by ** were retrieved by using the LIRIC algorithm. Only the values given for all the wavelengths refer to the column-integrated property. Day: AOD 440 nm $\approx 1.35 \pm 0.20$; $\alpha \approx -0.04 \pm 0.01$. Night: AOD 440 nm $\approx 0.83 \pm 0.03$; $\alpha \approx 0.08 \pm 0.02$.

| $\lambda$ [nm] | GARRLiC | | | | | AERONET | | | Raman (Day) | Raman + Regularization (Night) | | | |
|---|---|---|---|---|---|---|---|---|---|---|---|---|---|
| | $r_{eff}$ [$\mu m$] | Sph % | RRI | IRI | LR [sr] | RRI | IRI | LR [sr] | LR [sr] | $r_{eff}$ [$\mu m$] | RRI | IRI | LR [sr] |
| 355 | | | 1.59 | 0.003 | 37 | | | 82** | ~57 | | | | ~70 |
| 440 | | | 1.59 | 0.003 | 33 | 1.54±0.06 | 0.0045 | 74 | | | | | |
| 532 | | | 1.59 | 0.002 | 28 | | | 58** | ~53 | | | | ~58 |
| 675 | 1.9 | 20% | 1.58 | 0.002 | 25 | 1.53±0.05 | 0.0016 | 43 | | 1.1 | 1.53 | 0.010 | |
| 870 | | | 1.57 | 0.002 | 24 | 1.53±0.07 | 0.0011 | 37 | | | | | |
| 1020 | | | 1.56 | 0.002 | 22 | 1.53±0.07 | 0.0010 | 35 | | | | | |
| 1064 | | | 1.56 | 0.002 | 22 | | | 34** | | | | | |

**Table 2.** = Tabel 3 from article. Aerosol properties during the dust event over the Dakar site on 10 April 2015. The LR values marked by ** were retrieved by the LIRIC algorithm. Only the values given for all the wavelengths refer to the column-integrated property. Day: AOD 440 nm $\approx 1.53\pm0.04$; $\alpha \approx 0.02\pm0.01$. Night: AOD 532 nm $\approx 0.83$; $\alpha \approx 0$ by Raman.

| $\lambda\,[nm]$ | GARRLiC | | | | | AERONET | | | Raman (Day) | Raman + Regularization (Night) | | | |
|---|---|---|---|---|---|---|---|---|---|---|---|---|---|
| | $r_{eff}$ [$\mu m$] | Sph % | RRI | IRI | LR [sr] | RRI | IRI | LR [sr] | LR [sr] | $r_{eff}$ [$\mu m$] | RRI | IRI | LR [sr] |
| 355 | | | 1.60 | 0.004 | 20 | | | 70** | $\sim$25 | | | | $\sim$59 |
| 440 | | | 1.60 | 0.003 | 17 | 1.60±0.08 | 0.0058 | 62 | | | | | |
| 532 | | | 1.60 | 0.003 | 14 | | | 49** | $\sim$23 | | | | $\sim$50 |
| 675 | 2.0 | 57% | 1.60 | 0.002 | 13 | 1.60±0.05 | 0.0020 | 39 | | 0.9 | 1.54 | 0.008 | |
| 870 | | | 1.59 | 0.002 | 12 | 1.58±0.05 | 0.0014 | 32 | | | | | |
| 1020 | | | 1.58 | 0.002 | 13 | 1.58±0.06 | 0.0014 | 31 | | | | | |
| 1064 | | | 1.58 | 0.002 | 13 | | | 30** | | | | | |

[Figure]

**Figure 1.** = Figure 16 from article. Volume concentration profiles for an event over the Dakar site on 10 April 2015.